# Identification and Characterization of α-Actinin 1 of *Histomonas meleagridis* and Its Potential Vaccine Candidates against Histomonosis

**DOI:** 10.3390/ani13142330

**Published:** 2023-07-17

**Authors:** Dandan Liu, Chen Chen, Qiaoguang Chen, Shuang Wang, Zaifan Li, Jie Rong, Yuming Zhang, Zhaofeng Hou, Jianping Tao, Jinjun Xu

**Affiliations:** 1College of Veterinary Medicine, Yangzhou University, Yangzhou 225009, China; ddliu@yzu.edu.cn (D.L.); 13912133544@163.com (C.C.); chenqiaoguang@126.com (Q.C.); 15861321371@163.com (S.W.); lizaifan@163.com (Z.L.); rongjie950328@163.com (J.R.); 15725524962@163.com (Y.Z.); zfhou@yzu.edu.cn (Z.H.); yzjptao@126.com (J.T.); 2Jiangsu Co-Innovation Center for Prevention and Control of Important Animal Infectious Diseases and Zoonoses, Yangzhou University, Yangzhou 225009, China; 3Shanghai Wildlife and Protected Natural Areas Research Center, Shanghai 200366, China

**Keywords:** histomonosis, *Histomonas meleagridis*, α-actinin 1, prokaryotic expression, immunolocalization, immune protection

## Abstract

**Simple Summary:**

*Histomonas meleagridis* is a protozoan parasite that causes histomonosis in gallinaceous birds. This study identified α-actinin 1 of *H. meleagridis.* α-actinin 1 was predicted as a virulence factor involved in cell adhesion in our previous differential proteomic analysis. It localized in different cellular regions as detected by an indirect immunofluorescent assay between virulent histomonads and attenuated histomonads JSYZ. Administration of recombinant expressed rHma-actinin 1 reduced lesion scores in the cecum and liver, improved weight gain, and induced humoral and cellular immune responses against *H. meleagridis* in chickens.

**Abstract:**

*Histomonas meleagridis* is a protozoan parasite that causes histomonosis in gallinaceous birds such as turkeys and chickens. Since the banning and restricted usage of effective drugs such as nitarsone, 80–100% morbidity and mortality occur in turkeys and 20–30% mortality in chickens. New ideas are needed to resolve the re-emergence of histomonosis in poultry. In this study, the α-actinin encoding gene from *H. meleagridis* was cloned. The 1839-bp gene encoding 612 amnio acids showed close phylogenetic relationships with *Trichomonas vaginalis* and *Tritrichomonas foetus*. It was then inserted into the prokaryotic expression vector pET28a(+) and induced with isopropyl-β-D-thiogalactopyranoside. A 73 kDa recombinant protein rHmα-actinin 1 was obtained and purified with a Ni-NTA chromatography column. rHmα-actinin 1 was recognized by mouse anti-rHmα-actinin 1 polyclonal antibody, mouse anti-rHmα-actinin 1 monoclonal antibody, and rehabilitation sera from *H. meleagridis* infected chickens. Native α-actinin 1 in the total proteins of *H. meleagridis* can also be detected with mouse anti-rHmα-actinin monoclonal antibody. Immunolocalization assays showed that Hmα-actinin 1 was mainly distributed in the cytoplasm of virulent histomonads JSYZ-D9 and in the peripheral regions (near the plasma membrane) of attenuated histomonads JSYZ-D195. Based on *in vivo* experiment, when chickens were subcutaneously immunized with rHmα-actinin 1 at 5 and 12 days old and then challenged with *H. meleagridis* at 19 days old, rHmα-actinin 1 reduced the lesion scores 12 days after infection (31 days old) and increased the body weight gain during the challenged period (19–31 days old). Furthermore, it also strengthened the cellular and humoral immune responses 7 days after the second immunization (19 days old). In conclusion, Hmα-actinin 1 could be used as a candidate antigen to develop vaccines against chicken histomonosis.

## 1. Introduction

*Histomonas meleagridis* is a protozoan parasite belonging to the phylum Parabasalia, class Tritrichomonadea, order Tritrichomonadida, and family Dientamoebidae [1]. It causes histomonosis (e.g., blackhead disease or typhlohepatitis) of Galliformes and is characterized by yellowish diarrhea, caseous cecal cores, and yellowish-green volcanic vent shaped focal necrosis on the surface of liver. For public health reasons, many effective prophylactic and therapeutic drugs to treat histomonosis have been banned in many countries [2,3]. As a result, turkeys suffer from 80–100% morbidity and mortality [1,4]. Chickens have lower morbidity and mortality rates compared to turkeys [5] but they can reach 20–30% of mortality in eastern China [6]. Chicken is the main source of meat for the Chinese population, and there has been a gradual development of ecological free-range poultry in recent years [7]. Whereas the prevalence of *H. meleagridis* infection in free-range chickens is attributed to low biosecurity measures, which make it easier for the intermediate vector *H. gallinarum* to transmit the infection [8]. In the past, the use of chemicals significantly reduced the occurrence of histomonosis. Unfortunately, this disease has often been overlooked by farmers and veterinarians in China [6]. Experimental studies have demonstrated that histomonosis in layer chickens can lead to a decline in egg production by up to 30%, along with lower body weight gain and compromised eggshell quality [9,10]. New ideas and methods need to be explored to resolve the re-emergence of histomonosis in the poultry industry.

*In vitro* attenuated *H. meleagridis* successfully protects against the virulent strain in experimental conditions [9,11,12]. When 1-day-old turkeys were orally administrated attenuated *H. meleagridis* and challenged with the virulent strain, no negative effects were observed on growth performance in the vaccination phase compared to the non-challenged group, especially weight gain, which is the most important clinical parameter [11]. When 14-day-old turkeys were cloacally vaccinated with attenuated *H. meleagridis* and challenged with the virulent strain at 42 days of age, no obvious pathological lesions were observed at 28 days post infection (dpi), and no *H. meleagridis* DNA was detected in the liver. However, in the non-vaccinated but challenged group, initial severe clinical symptoms became apparent on 8 dpi. The first bird died on 11 dpi, and by 23 dpi, all the birds either died or exhibited poor condition [12]. Furthermore, vaccination of pullets with attenuated *H. meleagridis* can effectively prevent the drop in egg production of commercial layers in experimental conditions [9]. These successful vaccination experiments provide evidence for immunological methods to control histomonosis. However, attenuated *H. meleagridis* is not ready to be used as a commercial vaccine because of factors such as the complicated culture and storage conditions required *in vitro* and the risk of virulence return. Recombinant subunit vaccines are generally considered safe because they do not contain live pathogens. Their ability to specifically target antigens or molecular regions of a pathogen can potentially enhance their efficacy [13]. These vaccines can be produced using recombinant DNA technology, allowing for large-scale manufacturing [14]. However, recombinant subunit vaccines focus on specific antigens, they may not activate all aspects of the immune system, potentially leading to a weaker or shorter-lasting immune response [15,16]. Nonetheless, in the case of histomonosis prevention, recombinant subunit vaccines relying on the screening of antigenic genes may present an alternative choice. Subcutaneous administration of vaccines is commonly chosen due to its efficient absorption, slow and sustained release, ease of administration, reduced risk of injury, and suitability for a wide range of vaccines [13].

Many *H. meleagridis* specific proteins have been reported, including proteins involved in hydrogenosomal carbon metabolism [17], proteins homologous to intracellular and surface proteins [18], β-tubulin [19], hydrogenosomal proteins [20], potential antigenic α-actinin [20,21], and other cytoadherence and cellular stress management proteins [22].

Actin is common in the eukaryotic cell cytoskeleton. α-actinin is an actin binding protein and plays an important role in actin cross-link [23,24]. It is also involved in cell adhesion, cell motion, and morphological changes, as has been verified in *Trichomonas vaginalis* [13,25]. In *T. vaginalis*, α-actinin located in the cytoplasm when the parasite is pear-shape, whereas in the peripheral regions, cells adhere and become amoeboid, indicating transformation of *T. vaginalis* [25]. As a cell cytoskeleton-associated protein, α-actinin is considered a virulence factor involved in cell adhesion [26]. Although the cytoskeleton may be a potential pharmacological target [27], characterization analysis of α-actinin may provide new ideas to develop vaccines against histomonosis.

It has been reported that α-actinin of *T. vaginalis* is a common immunogen that induces host immune responses in serum of women exposed to *T. vaginalis* [28,29]. Recombinantly expressed α-actinin can also protect mice against *T. vaginalis* infection in experimental conditions [13]. There are three α-actinins in *H. meleagridis* [21]. Our previous transcriptome and proteomics studies on attenuated and virulent strains of *H. meleagridis* showed that α-actinin 1 was expressed at higher levels in virulent strains than attenuated strains [30]. In this study, α-actinin 1 gene (GenBank accession number: MW197428) was cloned and expressed for further research on its molecular characterization, localization in parasites, and potential immune protection properties against histomonosis.

## 2. Material and Methods

### 2.1. Animals

A total of 70 1-day-old specific pathogen free (SPF, with no infection with bacterium, parasites, and virus) layer type chickens (Leghorn, Jiangsu Lihua Animal Husbandry Co., Ltd., Changzhou, Jiangsu, China) were housed in cages (10 chickens per cage) in an *H. meleagridis*-free environment and fed *ad libitum* with food and water except for 6 h before and after challenge assays.

Six-week-old BALB/c mice (Experimental Animal Center of Yangzhou University, Yangzhou, Jiangsu, China) were housed in a pathogen-free environment with free access to water and food.

All animal procedures were performed following the guidelines and regulations of the Animal Experiment Ethics Committee of Yangzhou University with the license no. SYXK (SU) 2021-0027 for chickens and no. SYXK (SU) 2022-0044 for mice.

### 2.2. Parasites

The clonal culture of *H. meleagridis* JSYZ-D isolated from backyard flocks in Tianchang city (Anhui, China) was used in this study [30]. The virulent histomonads named JSYZ-D9 (*in vitro* for nine passages) were used for gene cloning, challenge, and indirect immunofluorescence assays, and the attenuated histomonads named JSYZ-D195 (*in vitro* for 195 passages) were used only for indirect immunofluorescence assays.

Virulent and attenuated histomonads (5 × 10^5^ cells in 1 mL culture freezing medium) were recovered from liquid nitrogen into 10 mL culture medium containing 80% (*v/v*) Medium 199 (Gibco^TM^, Thermo Fisher Scientific, Waltham, MA, USA), 20% (*v/v*) horse serum (Gibco^TM^), 10 mg rice starch (Sigma-Aldrich^®^, St. Louis, MO, USA), and proper cecal flora of chicken, and cultivated in anaerobic conditions at 42 °C for 3 to 5 days.

### 2.3. Gene Cloning and Sequence Analysis

Total RNA of *H. meleagridis* was extracted using a TaKaRa MiniBEST Universal RNA Extraction Kit (Takara Bio. Inc., Dalian, China) following the manufacturer’s instructions. cDNA was synthesized with a PrimeScript^TM^ RT reagent Kit with gDNA Eraser (Takara). The α-actinin 1 sequence was amplified using PrimeSTAR^®^ Max DNA Polymerase (Takara) following the manufacturer’s instruction. PCR reactions were performed with the forward primer 5’-CG*GGATCC*ATGACTATTCTTGATAAAGGATGGGAA-3’ (*Bam*H I restriction enzyme sites in italic) and the reverse primer 5’-ATAGTTTA*GCGGCCGC*TTAAGCGTAAATGCTTTCGACC-3’ (*Not* I restriction enzyme sites in italic) using the following cycling parameters: 95 °C for 5 min, followed by 30 cycles of 98 °C for 10 s, 56 °C for 30 s, 72 °C for 2 min, and 72 °C for 10 min.

The amplified α-actinin 1 was subcloned into pGEM^®^-T Easy vector (Promega Corp., Madison, WI, USA) for sequence analysis. The protein sequence was predicted using Lasergene 7.0 software (DNASTAR, Madison, WI, USA) and analyzed using BLASTP.

Phylogenetic analysis of the amino acid sequences of α-actinin 1 was performed with other α-actinin proteins from different protozoan parasites (sequences were showed in Appendix A). The phylogenetic trees were constructed using maximum likelihood (ML) and assessed by bootstrap resampling on 1000 replicates. The G + L model was used as best nucleotide substitution model.

### 2.4. Prokaryotic Expression

The amplified α-actinin 1 fragment was inserted into the prokaryotic expression vector pET28a(+) (Invitrogen, Carlsbad, CA, USA) using *Bam*H I and *Not* I restriction enzyme sites. The constructed prokaryotic expression plasmid was named pET28a(+)-Hmα-actinin 1. It was transformed to *Escherichia coli* BL21 (DE3) (Invitrogen) and induced with 0.6 mM isopropyl-β-D-thiogalactopyranoside (IPTG) for 4 h at 37 °C. The induced recombinant bacteria were boiled with 1× sodium dodecyl sulfate-polyacrylamide gel electrophoresis (SDS-PAGE) loading buffer (Takara) for 10 min before 12% SDS-PAGE followed by western blot with an anti-6×his monoclonal antibody as the primary antibody.

The induced recombinant bacteria were also sonicated (2 s/3 s, for 3 min depending on the volume of the bacteria) to harvest the pellets and supernatant for solubility analysis with 12% SDS-PAGE.

The expressed recombinant proteins (rHmα-actinin 1) were purified with a 6× his-tagged Ni-NTA chromatography column (GenScript, Piscataway, NJ, USA) with 250 mM imidazole (Sigma). The purified rHmα-actinin 1 was dialyzed in phosphate-buffered saline (PBS; pH 8.0) at 4 °C for 8 h to remove imidazole and other molecules. The purified proteins were concentrated to 2 mg/mL using polyethylene glycol (PEG8000) and stored at −80 °C for further use.

### 2.5. Polyclonal Antibody Preparation

Mouse anti-rHmα-actinin 1 polyclonal antibody was prepared as follows. Briefly, 20 μg rHmα-actinin 1 in 50 μL was fully mixed with an equal volume of QuickAntibody-Mouse3W adjuvant (Biodragon, Beijing, China) following the manufacturer’s instructions. Six-week-old BALB/c mice were intramuscularly immunized two times at two-week intervals with a 100 μL mixture. Mouse anti-rHmα-actinin 1 polyclonal antibody serum was separated from the blood by centrifugation at 2000× *g* for 15 min 1 week after the second immunization and stored at −80 °C. The antibody titer was detected using an indirect enzyme-linked immunosorbent assay (ELISA). Afterwards, 1 μg/well rHmα-actinin 1, serum diluted 1:400, and 1:5000 diluted horseradish peroxidase (HRP)-conjugated goat anti-mouse IgG antibody were used for the ELISA. Serum of unimmunized mice was used as the negative control.

### 2.6. Monoclonal Antibody Preparation

Monoclonal antibodies against rHmα-actinin 1 were prepared based on a previous study [31]. After two immunizations following polyclonal antibodies prepared for this study, mice with higher antibody titers were boosted once with no adjuvant one week later. Three days later, the spleen cells of the immunized mice were fused with myeloma SP2/0 cells. Hybridoma culture supernatants were examined using ELISA assays. Positive hybridoma cells stably secreting specific monoclonal antibodies were obtained after three subclones using the limiting dilution method and continuous passage of positive hybridoma cells. The subtypes of monoclonal antibodies were identified with a mouse Ig typing kit (Cat. No. BF06002; Biodragon).

### 2.7. Western Blot

The purified rHmα-actinin 1 was subjected to 12% SDS-PAGE and transferred to a nitrocellulose membrane (Merck, Darmstadt, Germany) for western blot as previously described [32]. Mouse anti-rHmα-actinin 1 polyclonal antibody (dilution, 1:400), monoclonal antibody (hybridoma culture supernatant; dilution; 1:20), and rehabilitation sera from *H. meleagridis* infected chickens (dilution, 1:200) were used as the primary antibodies. HRP-conjugated goat anti-mouse IgG (dilution, 1:5000; Kirkegaard and Perry Laboratories, Inc. (KPL), Gathersburg, MD, USA) and HRP-conjugated rabbit anti-chicken immunoglobulin G (IgG; dilution, 1:1000; GenScript) were used as secondary antibodies. The results were visualized with electrochemiluminescence (ECL) reagents using a Tanon 5200 Chemiluminescent Imaging System (Tanon, Shanghai, China). Negative sera from mice and chickens were used as negative controls.

Native whole proteins of *H. meleagridis* were prepared by grinding in liquid nitrogen and resolved over 12% SDS-PAGE following western blot analysis. Mouse anti-rHmα-actinin 1 monoclonal antibody (hybridoma culture supernatant, dilution, 1:20) was used as the primary antibody, and HRP-conjugated goat anti-mouse IgG (dilution, 1:5000; KPL) was used as the secondary antibody. At the same time, rHmα-actinin 1 was used as the control.

### 2.8. Indirect Immunofluorescence Assays

Indirect immunofluorescence analyses were performed as described previously [32]. Briefly, virulent histomonads JSYZ-D9 and attenuated histomonads JSYZ-D195 were coated on a slide and fixed with methanol (−20 °C). Mouse anti-rHmα-actinin 1 polyclonal antibody and monoclonal antibody were used as the primary antibody. Fluorescein isothiocyanate (FITC)-conjugated goat anti-mouse antibody IgG (dilution, 1:100; KPL) was used as the secondary antibody. Images were obtained using laser scanning confocal microscopy (LSCM) (Leica DM2500, Leica Microsystems GmbH, Wetzlar, Germany). Negative sera from mice and negative hybridoma culture supernatants were used as negative controls.

The final images were adjusted for brightness and contrast using Adobe Photoshop 2020 (Adobe Systems, San Jose, CA, USA).

### 2.9. Immunization and Challenge

The experimental design is shown in Table 1. Seventy chickens were randomly divided into seven groups at 5 days of age, including 200 µg, 150 µg, 100 µg, and 50 µg recombinant protein immunized groups referenced in a previous study [33], adjuvant controls, unimmunized and challenged controls (UC, positive controls), and unimmunized and unchallenged controls (UU, negative controls). On day 5, recombinant protein immunized groups were subcutaneously immunized with 0.1 mL recombinant protein (200 µg, 150 µg, 100 µg, or 50 µg) mixed in an equal volume of Freund’s complete adjuvant (FCA; Sigma). On day 12, booster immunization was performed as the same of primary immunization with Freund’s incomplete adjuvant (FIA; Sigma). Using the same dose for boosters ensures that the immune system is exposed to a sufficient level of antigens to reinforce and sustain the immune response over time. On day 19, all chickens except those in the UU group were challenged with 3 × 10^5^ *H. meleagridis* JSYZ-D9 strain in 300 µL culture medium per chicken by the cloacal route. More details on the immunization and challenge schedule are shown in Table 1.

### 2.10. Detection of Serum Antibody Levels

Indirect ELISA was used to measure the IgY antibody response to rHmα-actinin 1 by isolating sera from whole blood of 5-day-old chickens (pre-immunization), 12-day-old chickens (7 days after the first immunization), and 19-day-old chickens (7 days after the secondary immunization) as described in our previous study [32,33]. Briefly, 1 μg/well of rHmα-actinin 1 was coated on 96-well plates following 1:200 dilution of serum. The result was observed with HRP-conjugated goat anti-chicken IgY (H + L) antibody (1:20,000; GenScript). Five chickens were randomly selected in each group, and each sample was analyzed in triplicate.

### 2.11. Determination of Serum Cytokine Levels

The cytokines interleukin (IL)-2, IL-4, IL-10, and interferon (IFN)-γ in serum were detected using commercial IL and IFN-γ detection kits (MEIMIAN Bio., Yancheng, China) according to the manufacturer’s instructions. The serum samples were the same as those used to detect serum antibody levels, and each sample was analyzed in triplicate.

### 2.12. Evaluation of Immune Protection

The immune protection efficacy was evaluated by survival rate, body weight gain (BWG), and lesion scores of the liver and caecum. Survival rate (%) was calculated based on the survival of chickens in each group. The body weight of each group was measured three times, at 5 days, 19 days, and 31 days of age. BWG1 was the body weight gain between 5 and 19 days of age (immunization period), and BWG2 was between 19 and 31 days of age (challenge period). Relative body weight gain (RWG) (%) was calculated as follows: [BWG of the immunized, adjuvant, or UC groups/BWG of UU (negative control group)] × 100. On day 31 (12 days post challenge), chickens were euthanized, and the liver and caecum were taken for lesion score evaluation, which was graded from 0–4 following a previous study [5].

### 2.13. Statistical Analysis

The data were statistically analyzed using one-way ANOVA by SPSS 22.0 (IBM-SPSS, Inc., Chicago, IL, USA) with Duncan’s multiple range test. Data were expressed as mean ± S.D. value. Differences were considered significant at *p* < 0.05.

## 3. Results

### 3.1. Gene Cloning and Sequence Analysis

The 1839-bp α-actinin 1 was cloned from *H. meleagridis* and submitted to GenBank with the accession number MW197428 for nucleotides and QZM06938 for amino acids. It was a full open reading frame encoding 612 amino acids with no signal peptide. Its nucleotide had 99.8%, 53.7%, and 61.8% homology to α-actinin 1 (GenBank accession number FM200068), α-actinin 2 (GenBank accession number FM200071), and α-actinin 3 (GenBank accession number FM200072), respectively, from *H. meleagridis* in Austria [21]. It also showed low homology with *T. vaginalis*, from 52.2% to 53.9%. Phylogenetic analysis of the amino acid sequences showed that it was in the same clade as α-actinins from *H. meleagridis* and *Tritrichomonas foetus*, and in a sister clade with *T. vaginalis* (Figure 1). Briefly, *T. foetus* had close phylogenetic relationships with *H. meleagridid* based on small subunit rRNA and β-tubulin sequence analyses from previous research [34,35], and the same result was observed in this study. Sequence analysis verified that the α-actinin 1 gene obtained in this study was truly an α-actinin 1 gene and highly conserved with other *H. meleagridis* isolates.

### 3.2. Prokaryotic Expression

The 1839-bp α-actinin 1 gene was inserted into pET28a(+) (Invitrogen) and successfully expressed *in vitro* in soluble form with a molecular weight of 73 kDa. Western blot analysis with anti-6×his monoclonal antibody confirmed that the recombinant protein rHmα-actinin 1 was specifically expressed (Figure 2). It was purified using a Ni-NTA chromatography column and ToxinEraser^TM^ endotoxin removal kit (Genscript) for further analysis. The final concentration was 4 mg/100 mL BL21(DE3).

### 3.3. Western Blot

#### 3.3.1. Titer of Polyclonal Antibodies

The titer of mouse anti-rHmα-actinin 1 polyclonal antibody was determined to be 102,400, which was the highest dilution when the positive serum was 2.1 times greater than the negative serum at OD_450 nm_.

#### 3.3.2. IgG Subtype of Monoclonal Antibodies

Two stable hybridoma cells lines against rHmα-actinin 1, 4C2 and 2F1, were obtained and isotyped IgG1. The supernatant of the cell culture of 2F1 was collected to assess the native proteins of Hmα-actinin 1 in *H. meleagridis* using western blot and indirect immunofluorescence.

#### 3.3.3. Reactogenicity of Recombinant Protein (Polyclonal, Monoclonal, and Recovery Serum)

Western blot analysis showed that the recombinant protein rHmα-actinin 1 was recognized by mouse anti-rHmα-actinin 1 polyclonal antibody (Figure 3A), monoclonal antibody (Figure 3C line 11) and rehabilitation sera from *H. meleagridis* infected chickens (Figure 3B), which verified that rHmα-actinin 1 had good reactogenicity for further research.

#### 3.3.4. Native Protein Verified by Monoclonal Antibodies

Monoclonal antibodies were used to detect the native Hmα-actinin 1 protein in *H. meleagridis*. As result, a specific 73 kDa band was detected (Figure 3C line 1), which was the same as rHmα-actinin 1 (Figure 3C line 2).

Overall, the expressed recombinant protein rHmα-actinin 1 had good reactogenicity, and the native Hmα-actinin 1 protein could be detected by monoclonal antibodies, which verified that Hmα-actinin 1 truly exists in *H. meleagridis*.

### 3.4. Indirect Immunofluorescence

Indirect immunofluorescence assays were used to analyze the localization of Hmα-actinin 1 in virulent (JSYZ-D9) and attenuated (JSYZ-D195) histomonads. Mouse anti-rHmα-actinin 1 polyclonal antibody successfully recognized the native Hmα-actinin 1 of *H. meleagridis* (Figure 4). This showed that Hmα-actinin 1 is mainly distributed in the cytoplasm of virulent histomonads JSYZ-D9 (Figure 5A–F) but in the peripheral regions (near the plasma membrane) of attenuated histomonads JSYZ-D195 (Figure 5G–L). Mouse anti-rHmα-actinin 1 monoclonal antibody showed the same results as polyclonal antibodies (Figure 5).

### 3.5. Serum Antibody Levels

The serum antibodies of chickens in response to immunization with rHmα-actinin 1 are shown in Figure 6. Serum antibody levels increased in rHmα-actinin 1 immunized groups 7 days after the secondary immunization (19-day-old chickens) (*p* < 0.05), especially in the 150 µg group (*p* < 0.05).

### 3.6. Serum Cytokine Levels

IFN-γ, IL-2, IL-4, and IL-10 responses to rHmα-actinin 1 immunization are shown in Figure 7. Briefly, IFN-γand IL-2 showed an increasing trend, whereas IL-4 and IL-10 showed a decreasing trend in the immunized groups. At 7 days following primary immunization (12 days of age), IFN-γ and IL-2 in the 200 µg, 150 µg, and 100 µg groups were significantly higher than in the other groups (*p* < 0.05), whereas IL-4 and IL-10 were significantly lower in the 200 µg and 150 µg groups (*p* < 0.05). At 7 days following secondary immunization (19 days of age), IFN-γ and IL-2 were still significantly higher in the 200 µg, 150 µg, and 100 µg groups than in the other groups (*p* < 0.05), whereas IL-4 and IL-10 were still lower in immunized groups (*p* < 0.05).

Overall, there was no difference in the four serum cytokines the day before immunization (5 days of age). The group immunized with 200 μg rHmα-actinin 1 had the best cellular immunity, followed by the 150 μg group.

### 3.7. Protective Effect of rHmα-Actinin 1

Seven groups were designed to evaluate the protective effect of rHmα-actinin 1. No chickens died, and the survival rate of every group was 100% (Table 2). On days 5–19 (immunization period), the BWG1 of chickens was the same across groups (*p* > 0.05) (Table 2). On days 19–31 (challenge period), the BWG2 of the chickens in the 150 µg and 100 µg groups was significantly higher than that of the chickens in the UC group (positive control) (*p* < 0.05) but did not differ significantly from that of chickens in the UU group (negative control) (*p* > 0.05). The RWG-2 was highest (97.93%) in group 150 µg, followed by group 100 µg (93.34%) (Table 2). Liver lesions were slight and showed no significant difference in any group (*p* > 0.05) (Table 2). However, groups immunized with recombinant proteins had lower lesion scores, especially the 200 µg and 150 µg groups. The cecal lesion scores were similar to the liver lesion scores; scores were significantly lower in the 200 µg and 150 µg groups than the other groups (*p* < 0.05), and scores were lowest in the 150 µg group.

## 4. Discussion

Re-emergence of histomonosis resulted from the mandatory prohibition of chemical drugs to maintain food safety, making it is necessary to find more effective methods to control this disease. It has been reported that α-actinin in *T. vaginalis*, which contains α-actinin 2 (110 kDa) [27] and α-actinin 3 (135 kDa) [36,37], are conserved cytoskeletal proteins involved in motion and morphological changes in cell adhesion. Recombinant α-actinin also induced serum antibodies against *T. vaginalis* infection in mice under experimental conditions [13,29]. Thus, the α-actinin subunit may be a potential immunogen to develop vaccines against *T. vaginalis* infection.

There are three α-actinins in *H. meleagridis*, α-actinin 1, α-actinin 2, and α-actinin 3 [21]. The α-actinin 1 encoding gene was cloned in this study and was 1839 bp encoding a full open reading frame with 612 amino acids without a signal peptide. According to amino acid sequence analysis, it was in the same clade with α-actinin from *H. meleagridis* and *T. foetus*, which had close evolutionary relationships to each other [34,35], and was in a sister clade with *T. vaginalis*. Overall, α-actinin showed evolutionary conservation in trichomonas, and the α-actinin 1 encoding gene in *H. meleagridis* was successfully cloned in this study.

To reveal the localization of Hmα-actinin 1 in the parasite and its role in virulence, it was successfully expressed *in vitro* using the prokaryotic expression vector pET28a(+). The expressed rHmα-actinin 1 was 73 kDa and was specially recognized by mouse anti-rHmα-actinin 1 polyclonal antibodies, mouse anti-rHmα-actinin 1 monoclonal antibodies, and rehabilitation sera from *H. meleagridis* infected chickens. This showed that rHmα-actinin 1 had suitable immunoreactivity and was suitable for further research. Native α-actinin 1 was also detected with mouse anti-rHmα-actinin 1 monoclonal antibodies in extracted proteins of whole parasites using western blot. Immunolocalization results showed that Hmα-actinin 1 was located in the cytoplasm of virulent histomonads JSYZ-D9 and in the peripheral regions (near the plasma membrane) of attenuated histomonads JSYZ-D195. This result further verified the differential expression of α-actinin 1 reported in our previous differential proteomic analysis, where it was highly expressed in virulent histomonads and expressed at low levels in attenuated histomonads [30]. This result was similar to that of previous research on *H. meleagridis* and *T. vaginalis*, which showed that α-actinin was located in the cytoplasm when it is pear-shaped, but when parasites adhered to host cells and become amoeboid shape, it was localized in the periphery [21,25,38]. Thus, we conclude that Hmα-actinin 1 may be a virulence-associated protein and play an important role in cytoskeleton changes and adhesion to host cells.

Recombinant expression technology can be used as a tool for studying gene function. α-actinin was found in *T. vaginalis* and successfully expressed in pET32a system in *E. coli* BL21 (DE3) cells. α-actinin was primarily localized in the cytoplasm of *T. vaginalis* by using indirect immunofluorescence with the polyclonal antibodies against α-actinin [13]. Another *T. vaginalis* α-actinin, α-actinin 2 (Tvα-actinin 2), was also successfully expressed in pET21b expression plasmid and transferred into *E. coli* BL21 (DE3) cells. Immunofluorescence localization showed that during trophozoite development, Tvα-actinin 2 was mainly located in the cytoplasm. However, during amoeboid stage development, Tvα-actinin 2 was primarily found near the plasma membrane. It was observed that *T. vaginalis* undergoes a morphological transition to an amoeboid-like form during the invasive stage. This further confirmed that Tvα-actinin 2 is a virulence factor associated with invasion [26].

Recombinant expression technology enables the mass production of recombinant subunit vaccines, thereby reducing the cost and time required for preparing natural protein vaccines [14]. Recombinant subunit vaccines can target specific subunit antigens, stimulating the specific immune response. They exhibit higher safety levels, reducing the risk of reversion to virulence and dissemination of pathogens associated with live attenuated vaccines [14]. In this study, animal experiments were designed to evaluate the immune protective effect of rHmα-actinin 1 depending on mortality rate, lesion score, body weight gain, and cellular and humoral immune responses.

*In vivo* protective experiments showed that rHmα-actinin 1 immunized groups showed lower cecal and liver lesions and higher BWG than adjuvant and UC groups. No chickens died in the groups. The liver lesion score was slight compared to caecum because the characterization of the immune system in chickens prevents parasites transmitting from caecum to liver [39,40]. Liver score showed no difference in groups; however, there were significant differences in the number of chickens with liver lesions among each group. No visible liver lesions were observed in the 200 µg, 150 µg, and UU groups. One chicken each with a mild lesion (1 point) was in the 100 µg and 50 µg groups, respectively. Four chickens in the adjuvant group exhibited varying degrees of lesions, from 1 to 2 point. Two chickens in the UC group showed lesions, 2 and 1.5 point. On the body weight gain, there was no difference between the groups during immunization period (5–19 days of age). This indicated that immunization with recombinant protein did not negatively affect body weight gain (BWG1). However, when the chickens were challenged with *H. meleagridis*, there were variations in body weight gain (BWG2) during challenge period (19–31 days of age) among the groups. The 150 µg group showed the highest level of protection compared to the other immunized groups.

rHmα-actinin 1 induced better humoral and cellular immune mechanisms mediated by antibodies and cytokines. Serum antibody levels showed no obvious increase 7 days after the first immunization but increased significantly after the second immunization. This growth trend was also reflected in IL-2 levels. IL-2 is considered a T-cell growth factor that can promote the activation and proliferation of lymphocytes and assist B lymphocytes to produce specific antibodies against protozoan infection [5,41]. In this study, IL-2 showed a similar trend as serum antibody levels in chickens immunized with rHmα-actinin 1. IFN-γ levels were similar to IL-2, showing a trend of increasing with immunization time. IFN-γ and IL-2 are both proinflammatory cytokines. Immunized chickens can inhibit the migration of the parasite from the caecum to liver, leading to milder lesions in the liver. This result was similar to previous research that showed that chickens had less severe liver lesions than turkeys because of earlier cytokine expression profiles in the caecum [5]. IL-4 and IL-10 showed a decreasing trend in the immunized groups, especially the 200 µg and 150 µg groups. However, they did not react as similarly as IFN-γ and IL-2. IFN-γ and IL-2 are Th1-related cytokines, whereas IL-4 and IL-10 are Th2-related cytokines. It has been reported that *H. meleagridis* infection stimulates higher Th1 immune responses than Th2 [39,42].

α-actinin 1 was successfully identified in this study, but the molecular mechanism associated with its different expression and localization in virulent versus attenuated strains needs to be further researched. It is recommended to conduct a preliminary experiment before *in vivo* experiment to analyze the peak period of lesions after *H. meleagridis* infection, in order to determine the timepoint for necropsy.

## 5. Conclusions

In summary, this study cloned the 1839-bp α-actinin 1-encoding gene of *H. meleagridis*. It was successfully expressed in the prokaryotic expression vector pET28a(+) with a molecular weight of 73 kDa. rHmα-actinin 1 showed good immunogenicity and was detected by mouse anti-rHmα-actinin 1 polyclonal antibodies, mouse anti-rHmα-actinin 1 monoclonal antibodies, and rehabilitation sera from *H. meleagridis*-infected chickens. Immunolocalization analysis showed that α-actinin 1 localized to the cytoplasm of virulent histomonads and in the peripheral regions (near the plasma membrane) of attenuated histomonads. Results indicate that Hmα-actinin 1 may be an important virulence infector. The recombinant protein rHmα-actinin 1 can also stimulate humoral and cellular immune responses against *H. meleagridis* infection in chickens. Overall, Hmα-actinin 1 may be a potential vaccine candidate against chicken *H. meleagridis* infection.

## Figures and Tables

**Figure 1 animals-13-02330-f001:**
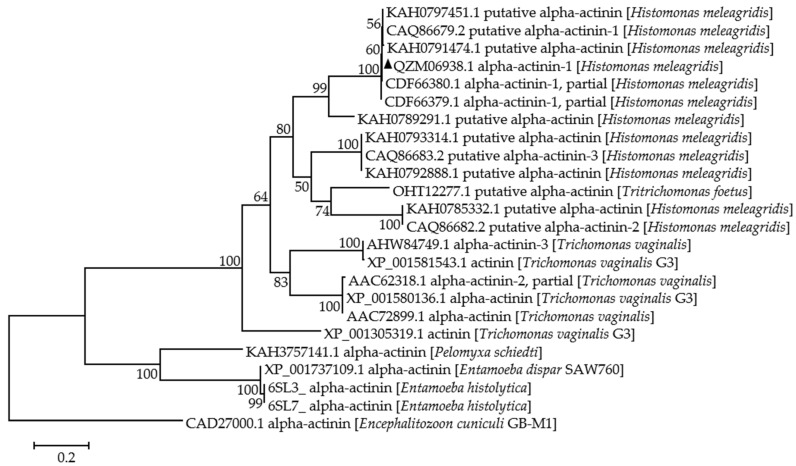
Phylogenetic analysis of the amino acid sequences of α-actinins from different protozoan parasites. Percentage support (>50%) from 1000 pseudoreplicates. Sequence labeled with black triangles was obtained in this study.

**Figure 2 animals-13-02330-f002:**
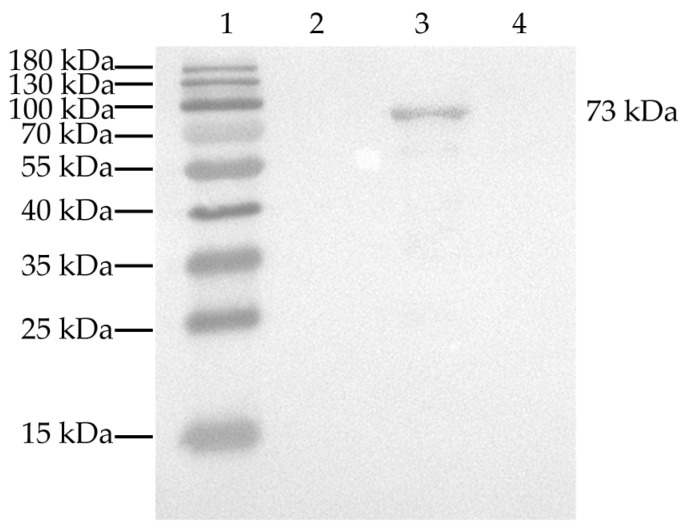
Western blot analysis of rHmα-actinin 1 by using anti-6×his monoclonal antibody. Line 1: Prestained protein molecular weight marker; 2: pET28a(+)/BL21 induced by IPTG; 3: pET28a(+)-Hmα-actinin 1/BL21 induced by IPTG; 4: BL21 induced by IPTG. IPTG: isopropyl-β-D-thiogalactopyranoside.

**Figure 3 animals-13-02330-f003:**
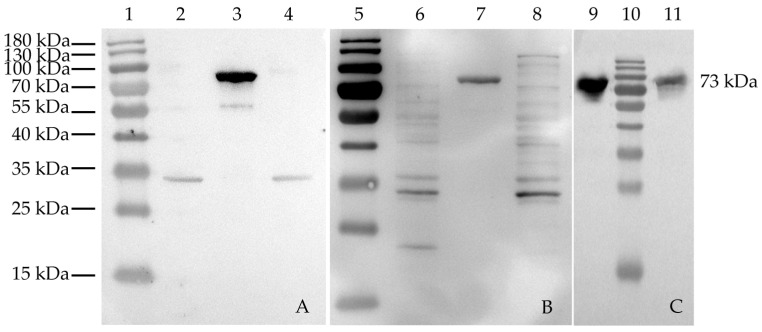
Western blot analysis of the reactogenicity of rHmα-actinin 1 recognized by mouse anti-rHmα-actinin 1 polyclonal antibody (**A**) and monoclonal antibody (**C**; line 11) and rehabilitation sera from *H. meleagridis* infection chickens (**B**), and native proteins of Hmα-actinin 1 recognized by monoclonal antibody (**C**; line 9). Line 1, 5, and 10: Prestained protein molecular weight marker; 2 and 6: pET28a(+)/BL21 induced by IPTG; 3 and 7: pET28a(+)-Hmα-actinin 1/BL21 induced by IPTG; 4 and 8: BL21 induced by IPTG; 9: native whole proteins extracted from *H. meleagridis*; 11: purified rHmα-actinin 1 protein.

**Figure 4 animals-13-02330-f004:**
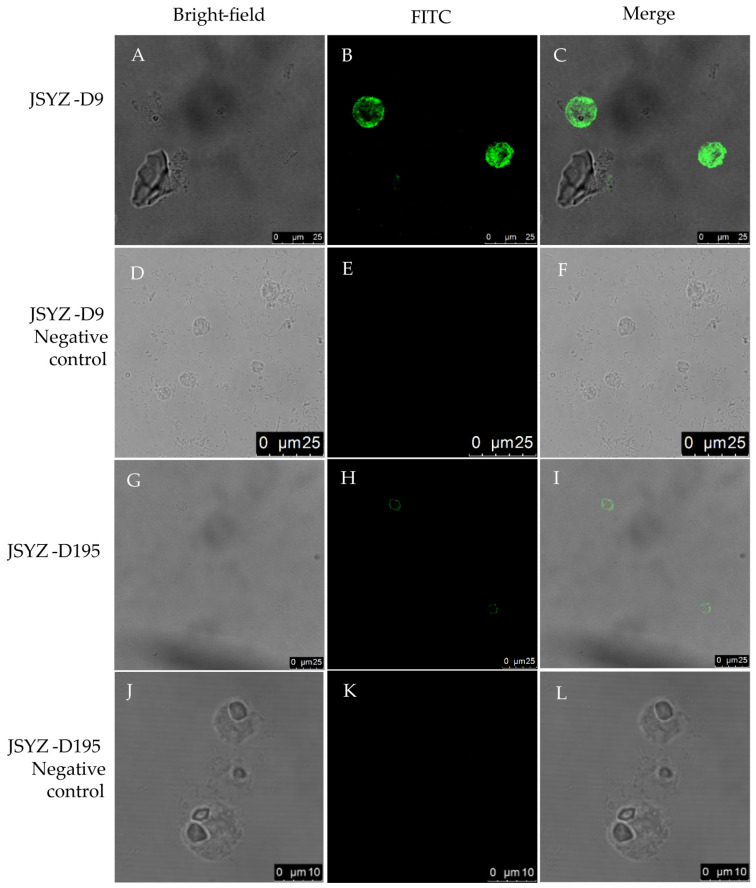
Indirect immunofluorescence localization of Hmα-actinin 1 in the virulent and attenuated histomonads with mouse anti-rHmα-actinin 1 polyclonal antibody. (**A**–**F**): virulent histomonads JSYZ-D9, Hmα-actinin 1 mainly distributed in the cytoplasm; (**G**–**L**): attenuated histomonads JSYZ-D195, Hmα-actinin 1 mainly distributed in the peripheral regions (near plasma membrane). (**D**–**F**) were JSYZ-D9 negative control; (**J**–**L**) were JSYZ-D195 negative control. First antibody was mouse anti-rHmα-actinin 1 polyclonal antibody, second antibody was FITC-conjugated goat anti-mouse IgG antibody.

**Figure 5 animals-13-02330-f005:**
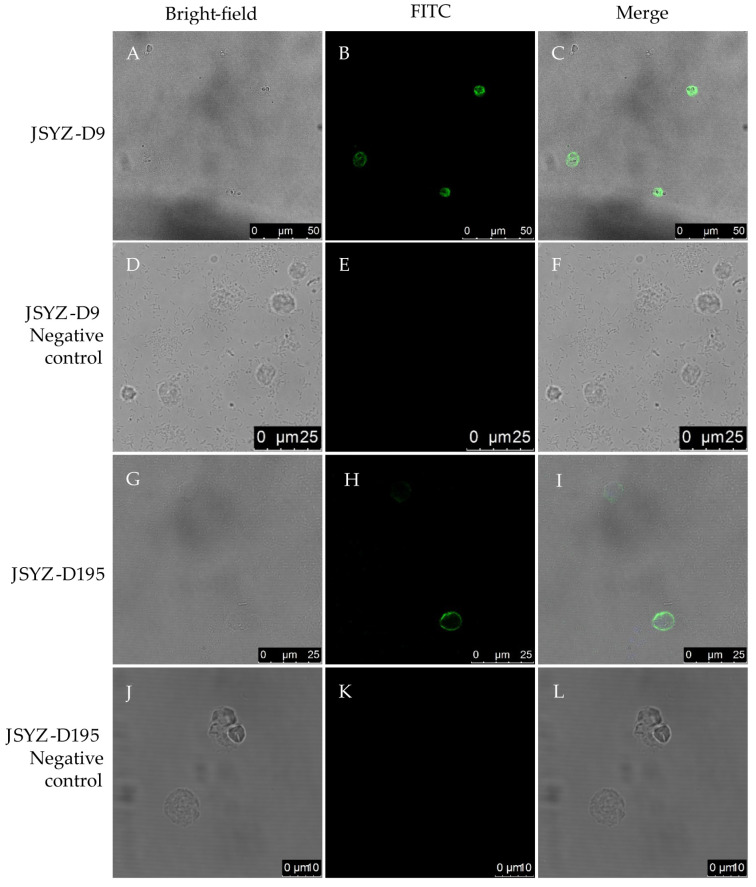
Indirect immunofluorescence localization of Hmα-actinin 1 in the virulent and attenuated histomonads with mouse anti-rHmα-actinin 1 monoclonal antibody. (**A**–**F**): virulent histomonads JSYZ-D9, Hmα-actinin 1 mainly distributed in the cytoplasm; (**G**–**L**): attenuated histomonads JSYZ-D195, Hmα-actinin 1 mainly distributed in the peripheral regions (near plasma membrane). (**D**–**F**) were JSYZ-D9 negative control; (**J**–**L**) were JSYZ-D195 negative control. First antibody was mouse anti-rHmα-actinin 1 monoclonal antibody, second antibody was FITC-conjugated goat anti-mouse IgG antibody.

**Figure 6 animals-13-02330-f006:**
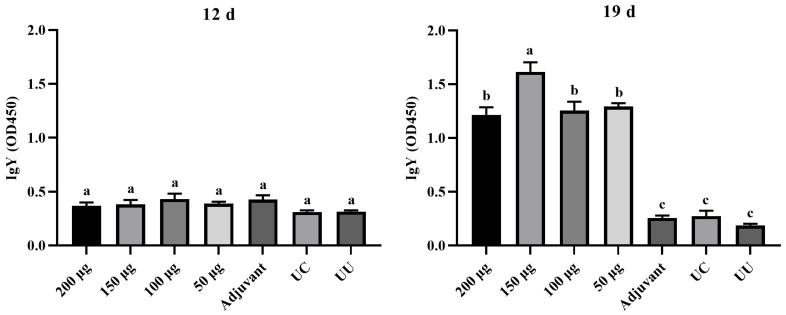
ELISA detection of serum antibody levels response to immunization with rHmα-actinin 1. The antibody levels increased in the groups 200, 150, 100, and 50 μg after the second immunization (19 days old), and were significantly higher than in the groups Adjuvant, UC, and UU (*p* < 0.05). Each bar represents the mean ± S.D. value (n = 5), and bars with different letters are significantly different (*p* < 0.05) according to the Duncan’s multiple range test. UC: unimmunized and challenged control (positive control); UU: unimmunized and unchallenged control (negative control).

**Figure 7 animals-13-02330-f007:**
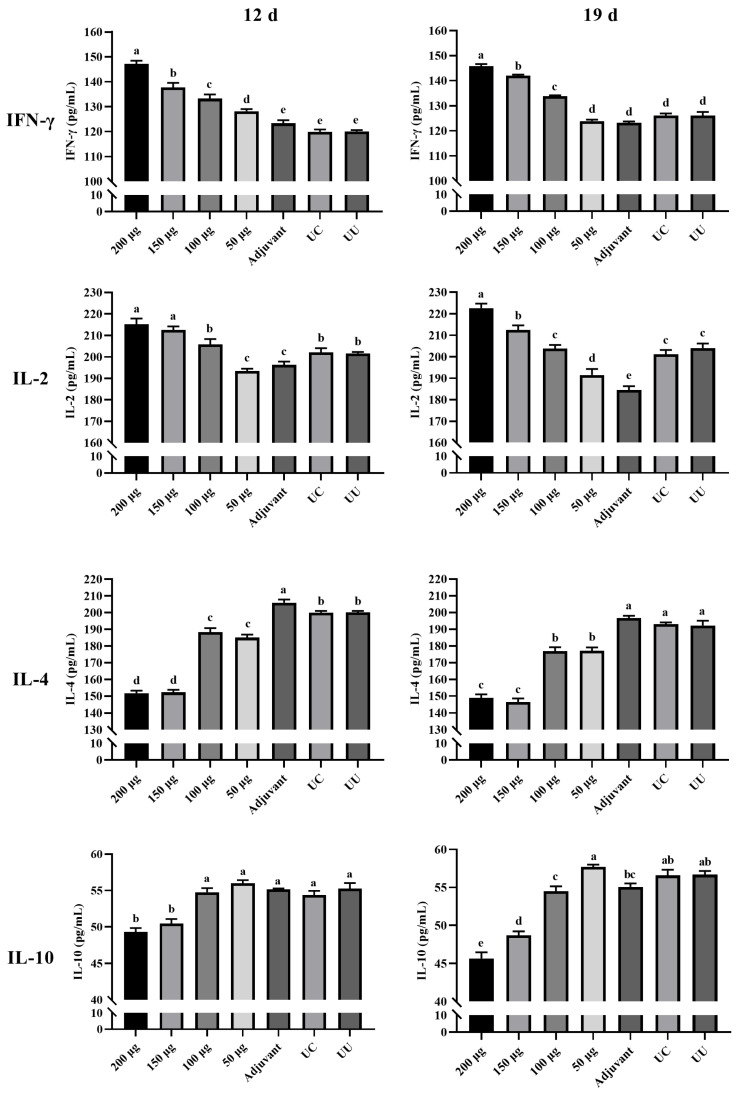
IFN-γ, IL-2, IL-4, and IL-10 responses to rHmα-actinin 1 immunization. IFN-γ, IL-2, IL-10, and IL-4 were detected by ELISA at 7 days following primary and secondary immunization. Each bar represents the mean ± S.D. value (n = 5), and bars with different letters are significantly different (*p* < 0.05) according to Duncan’s multiple range test. UC: unimmunized and challenged control (positive control); UU: unimmunized and unchallenged control (negative control).

**Table 1 animals-13-02330-t001:** Details of the immunization and challenge schedule.

Groups	No. of Chickens	Immunization—Dose and Route		Challenge—(Dose and Route)
		5-Day-Old Chicken	12-Day-Old Chicken	19-Day-Old Chicken
200 μg	10	rHmα-actinin1 200 μg S.C route * Adjuvant: FCA **	rHmα-actinin1 200 μg S.C route Adjuvant: FIA ***	300,000 *H. meleagridis* JSYZ-D9 CR ****
150 μg	10	rHmα-actinin1 150 μg S.C route Adjuvant: FCA	rHmα-actinin1 150 μg S.C route Adjuvant: FIA	300,000 *H. meleagridis* JSYZ-D9 CR
100 μg	10	rHmα-actinin1 100 μg S.C route Adjuvant: FCA	rHmα-actinin1 100 μg S.C route Adjuvant: FIA	300,000 *H. meleagridis* JSYZ-D9 CR
50 μg	10	rHmα-actinin1 50 μg S.C route Adjuvant: FCA	rHmα-actinin1 50 μg S.C route Adjuvant: FIA	300,000 *H. meleagridis* JSYZ-D9 CR
Adjuvant	10	FCA plus PBS S.C route	FIA plus PBS S.C route	300,000 *H. meleagridis* JSYZ-D9 CR
UC	10	PBS via S.C route	PBS via S.C route	300,000 *H. meleagridis* JSYZ-D9 CR
UU	10	PBS via S.C route	PBS via S.C route	culture medium via CR

S.C *: Subcutaneous injection route; FCA **: Freund’s complete adjuvant; FIA ***: Freund’s incomplete adjuvant; CR ****: cloacal route; UC: unimmunized and challenged control (positive control); UU: unimmunized and unchallenged control (negative control).

**Table 2 animals-13-02330-t002:** Protective effect of rHmα-actinin 1.

Groups	Survival Rate (%)	Body Weight Gain				Average Lesion Score	
		BWG1 (g)	RWG1 (%)	BWG2 (g)	RWG2 (%)	Liver	Cecal
200 μg	100	112.23 ± 2.40 ^a^	93.56	80.25 ± 6.42 ^ab^	87.32	0.00 ± 0.00 ^a^	2.65 ± 0.33 ^c^
150 μg	100	120.67 ± 4.05 ^a^	100.59	90.00 ± 4.02 ^a^	97.93	0.00 ± 0.00 ^a^	1.30 ± 0.26 ^d^
100 μg	100	106.72 ± 3.10 ^a^	88.96	85.78 ± 4.46 ^ab^	93.34	0.20 ± 0.20 ^a^	3.00 ± 0.22 ^bc^
50 μg	100	119.58 ± 6.09 ^a^	99.68	61.53 ± 4.11 ^c^	66.95	0.20 ± 0.20 ^a^	3.45 ± 0.16 ^ab^
Adjuvant	100	107.67 ± 3.52 ^a^	89.75	75.26 ± 3.23 ^b^	81.89	0.65 ± 0.28 ^a^	3.80 ± 0.11 ^a^
UC	100	107.65 ± 3.17 ^a^	89.74	58.97 ± 6.47 ^c^	64.17	0.35 ± 0.24 ^a^	3.70 ± 0.15 ^a^
UU	100	119.96 ± 2.61 ^a^	100.00	91.90 ± 1.88 ^a^	100.00	0.00 ± 0.00 ^a^	0.00 ± 0.00 ^e^

BWG: body weight gain; RWG: relative body weight gain. BWG1 and RWG1 come from the body weight gain between 5 and 19 days (immunization period); BWG2 and RWG2 was between 19 and 31 days (challenge period). UC: unimmunized and challenged control (positive control); UU: unimmunized and unchallenged control (negative control). Data were expressed as mean ± S.D. value (n = 10). Results within a column with no common letter differ significantly at *p* < 0.05 according to Duncan’s multiple range test.

## Data Availability

The sequences presented in phylogenetic tree are available in Appendix A. Other supplementary data presented in this study are available on request from the corresponding author.

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
