# Peer review of "Identification and Characterization of α-Actinin 1 of *Histomonas meleagridis* and Its Potential Vaccine Candidates against Histomonosis"

_animals, 2023, doi:10.3390/ani13142330_

Round 1
Reviewer 1 Report
Comments:
-Overview of the manuscript: The manuscript focused on the evaluation of a novel vaccine against Histomonas meleagridis to reduce/prevent histomonosis in chickens, which is a disease of concern for the poultry industry. The vaccine was produced using a recombinant expression system for the α-actinin 1 gene of H. meleagridis, which was predicted from the group’s previous work to be a virulence gene. The vaccine was characterized and then administered using a subcutaneous injection approach. After vaccination, the chickens were challenged via the cloacal route with a virulent strain of H. meleagridis JSYZ-D9. The experimental design included seven groups of birds (N=10 birds/group); negative control, challenged, vaccinated (i.e., four individual groups with different vaccination amounts), and vaccinated plus challenged. A subset of birds (N=5 birds/group) were tested for response to vaccination on days 12 and 19. Then all birds were necropsied on day 31 of life which corresponds to 12 days post challenge. The vaccine effectiveness was evaluated for; 1. Serum antibody levels, 2. Serum cytokine levels, 3. Body weight gain, and 4. Lesion scores of liver and cecum.
-Major Strengths: The vaccine was novel (i.e., α-actinin 1 gene) and is a strength of the study. The experimental design (i.e., factorial and various assays) was also a strength of the study. The manuscript was generally well written. The non-challenged birds did not have lesions in any of the tissues, the vaccine reduced lesion scores in the liver of challenged birds (Table 2) and increased weight gain compared to unvaccinated challenged birds. The vaccination and/or challenge induced some significant (P<0.05) serum antibody levels and serum cytokines.
-Major Limitations: There are two major limitations of the study; 1. The statistical analyses need improvement, and 2. The discussion needs improvement. Additional limitations are provided as point-by-point responses below.
-Point-by-point revisions:
-Line 19: Remove “differentially expressed” and replace with “localized in different cellular regions as detected by an indirect immunofluorescent assay”. The term differentially expressed typically refers to differences in RNA expression levels, which this study did not assess.
-Line 21: Remove “liver”. This study did not identify any significant differences in lesion scores in the liver only in the cecum (Table 2).
-Line 37: Again, remove “liver” per above comment.
-Line 49: Add in sentence about the prevalence of histomonosis in chickens with appropriate literature reference.
-Line 69: Add in sentence about the pros and cons of using recombinant antigen vaccines. Add in the routes of administration for these vaccines and why you chose subcutaneous in your study.
-Line 76: alpha actin 1 proteins are found across various species. Add in a reference on this gene in chickens. Discuss potential limitations of using a vaccine that may be similar to a host gene. Later points on the phylogenetic analysis of this gene.
-Line 90: Include the unpublished data in this study or remove the reference to “data not published”.
-Line 97: Include the number of chickens per cage.
-Line 108: Add reference for this strain.
-Line 155: Revise sentence “centrifugation at 2500 r/min for 15 min 1 week later and 155 stored at -80°C.” As written it appears the serum was kept in the lab for 1 week.
-Line 206-207: provide justification for the recombinant protein dosage amounts. Was a pilot study done to choose these amounts or are they based on the literature? Provide justification for why the booster amount was the same as the original vaccination amount. Often, boosters are half the original dose to prevent overstimulation of the immune system.
-Table 1 final line: Provide justification for why the UU (unchallenged unvaccinated) group was not given a mock challenge. There is a line in the table. Unchallenged animals used as a control should be subject to the same experimental handling methods as challenged animals.
-Line 223: Provide your method for choosing birds randomly. Simply catching the first 5 birds one can is not random. Include if the birds had individual identification numbers or wing bands.
-Line 233: Body weight is extremely variable at the individual animal level. The body weight gain should be assessed per animal in each group then provide an average with standard deviation for that group. Then that can be compared to the average of the control groups. Because of rapid growth, the day of measurement is critically important to the evaluation of this data. Break down the body weight measurements for each day rather than a span as you indicate “BWG was first measured between 5–19 days of 233 age (immunization period), then between 20–31 days of age (challenge period)”. Include which exact days the animals were measured for body weight. Body weight of layers double between 5 days and 19 days of age making the interpretation of this data impossible.
-Line 240: This is insufficient for the various types of data that were analyzed. Include information about the method used for triplicate analysis used for Elisa and cytokine expression. Were outliers discarded? Was the mean value used among the triplicate? Was multiple testing correction employed, and if so what method was used?
-Line 250: This is first use of “phylogenetic analysis” in the main body of the manuscript. The methods must be included for the phylogenetic analysis in the Methods portion of the manuscript. To include enough detail to replicate the analysis. Sequences used to run the analysis must be clearly referenced and included as a supplementary file. In addition, chicken alpha actin 1 should be included in the phylogenetic analysis to identify any sequence similarity that exists between chicken and Histomonas meleagridis that could negatively impact the utility of the proposed vaccine due to negative impacts on the host cells.
-Figure 2: Spell out acronyms in figure legend.
-Figure 3: Revise the line numbers to range from 1-11 as the setup is currently confusing.
-Figure 5: Panel D “control” is cut off. Fix this.
-Figure 6: Why did you not test serum antibody levels in challenged birds at day of necropsy? Justification is needed in the Methods section of the manuscript for this. Line 334 remove “obviously”. Make y axes of each figure panel the same. Include the use of triplicates for each sample. Was the standard deviation of the triplicates included? The statistics need improvement. The legend says “bars with different letters are significantly different” but there are no letters. Include letters above bars. Multiple testing correction method needs to be included in the Methods section of the manuscript. The asterixis need to be explained or they should be replaced with letters to indicate significance. Why are the Adjuvant, UC, and UU not significantly different? The data do not make sense as presented.
-Figure 7. Same comments on the statistical description and analysis as Figure 6. The lines with 4 asterixis are very confusing and uninformative.
-Table 2: BWG1 and BWG2 need to be explained. This should be converted to body weight by day of measurement to be more informative. Include body weight at necropsy if possible. Was this a range in measurements as indicated in the Methods? If so, the results need to clearly indicate the days of measurements and what the comparisons are. Refer to comment in the Methods section about improving reporting for Body Weight traits. Include number of animals measured for each category on each day. Include statistical method for presumably multiple testing correction.
-Line 407: The discussion needs to be improved. Especially adding the pros and cons of using recombinant antigen vaccines vs other vaccination methods. The potential similarity of alpha actin 1 to host proteins needs to be addressed.
-Line 419: This statement indicates that the vaccine tested here reduced liver lesions. The data here do not support this conclusion. Liver lesions were not reduced significantly (Table 2).
-General discussion: The discussion needs improvement. Particularly the results need to be better interpreted within the context of the existing literature. The differences between vaccine dosage amounts needs to be addressed and discussed more thoroughly. The body weight gain data and lesion scores must be addressed in more depth. The serum antibody levels, and cytokine levels must be addressed and interpreted more fully. The limitations of the study (e.g., a single timepoint for necropsy) must be included. This will help the reader understand the significance and limitations of the current study and improve impact.
-Line 454: Data Availability Statement that says, “Not applicable”. Phenotypic data on each animal should be published as a supplemental file. The current study only summarizes the data with averages. Include the individual bird body weights, serum cytokine and antibody levels, and lesion scores.
English is generally good.
Author Response
Many thanks for your comments. The manuscript has been revised based on your comments. Please the attachment.

Reviewer 2 Report
This manuscript showed that Hmα-actinin 1 may be used as a candidate antigen to develop vaccines against chicken histomonosis. This manuscript has novel ideas and is well-written. However, it has some minor or major errors in the data interpretation for animal study. I hope the authors can address the issues.
Line 19. Remove “and”
Simple summary and abstract: I recommend not to use “It” “We” in the simple summary. Readers do not know what it is. Please clarify all of them.
Line 20: Remove “In this study”
Line 20: I do not think “Recombinant expressed rHma-actinin 1” reduced. It is a vaccine. So I suggest “Administration of recombinant expressed rHma-actinin 1 reduced lesion scores in the cecum and liver, improved weight gain and induced humoral and cellular immune responses against H. meleagridis in chickens.
Line 36: In vivo experiment
Line 37: I suggest Administration of rHmα-actinin 1 vaccines? At what age? What about inoculation of H. meleagridis? When did you measure lesion score, weight gain, and immune response?
Line 39: In conclusion, Hmα-actinin 1 could be used as a candidate antigen to develop vaccines against chicken histomonosis
Line 45: Kingdom: Protozoa; Phylum: Metamonada; Class: Parabasalia; Order: Trichomonadida; Family: Monocercomonadidae; Species: Histomonas meleagridis Please check the classification again.
Line 47: syn. -> e.g.,
Line 48: caseous cecal cores, and yellowish-green
Line 53: 20 – 30 % of what? Mortality?
Line 52: How about the production efficiency (e.g., growth performance, gut health, etc.)? In many articles, chickens only have asymmetric symptoms and just a carrier for H. meleagridis. I hope you can provide more logics why you wanted to study H. meleagridis in chickens.
Line 55: in which study? Please provide the references..
Line 57:no negative effects were observed on performance in the vaccination phase compared to the non-challenged group
Line 58: Growth performance
Line 61: Is 28 dpi peak time point? Please describe if there were lesion at the peak point (14 dpi).
Line 70: Many H. meleagridis specific proteins have been reported,
Line 96: What is specific pathogen free ? free from the H. meleagridis
Line 155: r/min? You meant rpm? If you want to use rpm, please provide the centrifuge machine information or use “g”
1 week late? What do you mean by 1 week later?
Line 157: Then -> afterwards
Line 174 : (hybridoma culture supernatant; dilution; 1:20)
Line 209: How much did you administrated ? 1 mL?
Line 240: Please write Duncan analysis.
Figure 6: Was it possible to calculate exact concentrations instead of OD450?
Figure 7: Why did not you use superscripts ? And why did you use Duncan instead of
Table 2: Please write statistical analysis. Please write the period BWG1 and BWG2. Why the liver had low lesion score? Do you think the challenge model was appropriate?
Discussion: Could you please add more references in the discussion? Some of the facts from the other papers do not have references.
Line 395 good -> suitable
Line 418: Can you say that in this study? The lesion score of liver was not high in the challenged control group. Please use the reference or delete it.
Line 423: What do you mean by strong? Please provide % or other way to justify “Strong”
It was good. Just minor issues.
Author Response

(The authors gave the same response as above.)

Round 2
Reviewer 2 Report
In vitro and in vivo should be italic